# Reliability of 3D Depth Motion Sensors for Capturing Upper Body Motions and Assessing the Quality of Wheelchair Transfers

**DOI:** 10.3390/s22134977

**Published:** 2022-06-30

**Authors:** Alicia Marie Koontz, Ahlad Neti, Cheng-Shiu Chung, Nithin Ayiluri, Brooke A. Slavens, Celia Genevieve Davis, Lin Wei

**Affiliations:** 1Human Engineering Research Laboratories, VA Pittsburgh Healthcare System, Pittsburgh, PA 15206, USA; neti.ahlad@pitt.edu (A.N.); joshua.chung.cs@pitt.edu (C.-S.C.); nia53@pitt.edu (N.A.); cgd16@pitt.edu (C.G.D.); liw49@pitt.edu (L.W.); 2Department of Bioengineering, Swanson School of Engineering, University of Pittsburgh, Pittsburgh, PA 15213, USA; 3Department of Rehabilitation Science and Technology, School of Health and Rehabilitation Sciences, University of Pittsburgh, Pittsburgh, PA 15213, USA; 4Collage of Health Sciences, University of Wisconsin-Milwaukee, Milwaukee, WI 53211, USA; slavens@uwm.edu; 5Texas Health Resources, Allen, TX 75013, USA

**Keywords:** skeletal tracking, depth sensor, machine learning, activities of daily living, biomechanics

## Abstract

Wheelchair users must use proper technique when performing sitting-pivot-transfers (SPTs) to prevent upper extremity pain and discomfort. Current methods to analyze the quality of SPTs include the TransKinect, a combination of machine learning (ML) models, and the Transfer Assessment Instrument (TAI), to automatically score the quality of a transfer using Microsoft Kinect V2. With the discontinuation of the V2, there is a necessity to determine the compatibility of other commercial sensors. The Intel RealSense D435 and the Microsoft Kinect Azure were compared against the V2 for inter- and intra-sensor reliability. A secondary analysis with the Azure was also performed to analyze its performance with the existing ML models used to predict transfer quality. The intra- and inter-sensor reliability was higher for the Azure and V2 (*n* = 7; ICC = 0.63 to 0.92) than the RealSense and V2 (*n* = 30; ICC = 0.13 to 0.7) for four key features. Additionally, the V2 and the Azure both showed high agreement with each other on the ML outcomes but not against a ground truth. Therefore, the ML models may need to be retrained ideally with the Azure, as it was found to be a more reliable and robust sensor for tracking wheelchair transfers in comparison to the V2.

## 1. Introduction

In 2016 there were approximately 282,000 individuals in the United States living with a spinal cord injury (SCI), with around 12,500 new cases occurring each year [1]. Individuals with SCI make up a portion of the over 3.6 million Americans over the age of 15 who use a wheelchair in their daily life as their primary means of mobility [2]. As a direct result of SCI or other conditions, wheelchair users (WUs) rely a great deal on their upper extremities, specifically the shoulders, arms, and the trunk, to complete common but essential activities of daily living (ADLs). This involves tasks such as, but not limited to: pushing their wheelchair, getting in and out of bed, cooking, cleaning, hygiene tasks, and performing transfers to a toilet, shower chair, or car. Manual wheelchair users (MWUs) will, on average, perform between 14–18 transfers each day; transfers are very physically demanding and may lead to and cause various forms of upper extremity pain and discomfort [3,4]. This is supported by research findings that 31–73% of MWUs have upper extremity pain, specifically in the shoulders [5]. This pain and discomfort force many MWUs to lead sedentary lifestyles and have a decreased quality of life [6].

Proper sitting-pivot-technique (SPT) for independent transfers in MWUs is vital to avoid predisposition to upper extremity pain and discomfort; therefore, proper training, practice, and management of SPTs are integral to the quality of life and opportunities for MWUs [4,7]. Methods to analyze the quality of SPTs include the Transfer Assessment Instrument (TAI), a reliable and validated tool that monitors the biomechanical properties of an individual while they perform an SPT [8]. A higher score on the TAI is indicative of better technique and lower mechanical loading on the upper extremity joints. The TAI includes components such as setup of wheelchair angle, hand grip, and body positioning. Proper utilization of the TAI has been shown to lead to better technique and biomechanics [9,10]; however, there is a heavy reliance on clinician and user education, training on proper technique, and subjective analysis of movement strategies [9,11]. Providing a system that could aid in the automatic assessment of proper versus improper transfer technique could increase TAI’s utility and the number of patients who could benefit from such an evaluation.

A large number of movement recognition and auto-technique classification applications have been described in the literature. These applications use a wide variety of sensors to capture the spatial and temporal features of the motions of interest. One group of applications uses inertial measurement units (IMUs) for capturing the biomechanical features of interest, which are used to classify the techniques [12,13,14]. Although these applications provide a wireless solution, the setup entails the attachment of physical sensors to the body as well as calibration procedures. Another group of applications uses an RGB camera and deep learning techniques to identify body keypoints and classify human activity [15,16]. While this type of application is inexpensive and non-intrusive, real-time 3D body point tracking is computationally intense and may not yet be accurate enough to allow for discerning proper from improper transfer technique using only 2D planar images [17,18].

A third group of applications entails the use of depth sensors which can provide full-body three-dimensional (3D) motion capture with facial, hand/finger, and/or body joint tracking capabilities while permitting total freedom of movement without holding or wearing any sensors or markers. These sensors are also inexpensive (<USD 500), portable, and easy to set up and use, making them an attractive option to consider for technical and clinical applications. Ensuring the simplicity of system use and flexibility of system setup is crucial to fit the needs of busy clinical professionals and to support the integration of an auto-scoring system into the existing facilities and treatment processes. The Microsoft Kinect V2 camera is a popular depth sensor that has been used in numerous studies to successfully automate the scoring of various clinical tools using 3D body tracking (skeletal) data [19,20,21].

Building upon the success of these prior studies, we developed TransKinect—a software application used to make TAI scoring easier and less subjective, using a Microsoft Kinect V2 depth camera to watch and auto-score the TAI. This is accomplished through trained machine learning (ML) models that use biomechanical features to determine if the SPT was conducted correctly or incorrectly in accordance with TAI principles [11]. The TransKinect application uses the sensor data to compute a large number of features, including max, min, range of motion, and averages of joint angles, rotations, displacements, velocities, accelerations, and jerk. These features, along with the ML models, are used to auto-score each item of the TAI as well as compute a total overall technique score [11]. The models used in the current version of TransKinect were trained from data collected from the Kinect V2; however, this sensor has since been discontinued.

Our research team has demonstrated that the Kinect V2 has the ability to recognize at-risk body motions during independent sitting pivot transfers [11,22,23]. The Intel RealSense D435 depth sensor could be a solution for Kinect V2’s discontinuation because its technical specifications are similar, and there is a full-body tracking software development kit (SDK) that provides similar 3D joint center locations as the V2. Similarly, V2’s successor, the Microsoft Azure, and a full-body skeletal tracking library in Python have recently become available.

For the continued development of TransKinect and successful translation into clinical practice, a new sensor must be identified. While there has been some research in comparing the performance of various depth sensors in their ability to track biomechanical variables [24,25,26,27], none of the studies that we know of have focused on motions that occur while in a seated position and for wheelchair transfer techniques specifically.

The overall goal of this study was to determine the potential for the RealSense and Azure sensors to act as surrogate sensors for the TransKinect application. Specifically, we aimed to determine the intra-rater (within sensor) reliability, inter-rater (between sensor) reliability, and agreement of select key features between the Intel RealSense and V2 and the Azure and V2. We hypothesized that the reliability within each sensor and between each sensor pairing would be high (interclass correlation coefficients (ICC > 0.80) and that the reliability would be similar between the sensors (ICC differences < 0.10). We further expected high agreement between the V2 and RealSense and V2 and Azure features (i.e., 95% of data falling within the mean ± (1.96 × STD) limits of agreement). A secondary goal of the study was to determine the agreement between TAI scores auto-generated with the V2 to those auto-generated with the Azure for proper and improper SPTs. We hoped to find that the Azure TAI scores would have high agreement with the V2 TAI scores (>70%) and that the Azure TAI scores would also have high accuracy with the ground truth scores (>70%).

## 2. Materials and Methods

Both phases of the study followed the proposed flowchart shown in Figure 1. The second phase had an additional analysis step involved for the machine learning.

The first phase of the study comparing the RealSense to V2 sensors was conducted pre-COVID-19 pandemic at the 2019 Annual National Veterans Wheelchair Games (NVWG) in Louisville, Kentucky. The second phase comparing the Azure and V2 sensors was conducted during the COVID-19 pandemic in the Human Engineering Research Laboratories.

### 2.1. Participants

The inclusion criteria for Phase 1 (RealSense vs. V2) participants were: (1) have a discernable neurological impairment affecting both lower extremities or persons with transfemoral or transtibial amputation of both lower extremities and who do not use prostheses during transfers, (2) at least one-year post-injury or diagnosis, (3) able to independently transfer to/from a wheelchair without human assistance or assistive devices, (4) use a wheelchair for the majority of mobility (over 40 h/week), and (5) over the age of 18 years. Participants were excluded if they had (1) current or recent history of pressure ulcers in the last year, (2) history of seizures or angina, or (3) were able to stand unsupported.

The Phase 2 (Azure vs. V2) participants were able-bodied research staff without any upper or lower limb impairment who were knowledgeable on proper and improper wheelchair transfer techniques.

For Phase 1 RealSense vs. V2 reliability and agreement analysis, 26 men and 4 women with an average age of 56.6 years (standard deviation (STD) = 11.8) contributed a total of 150 transfer trials for the analysis. The group regularly performed, on average, 13 transfers (STD = 10.9, self-reported) per day. Participants had 16.8 years (STD = 5.8) experience in using wheelchairs and used their wheelchairs for 13.2 h (STD = 5.8) per day. Nine (30%) were African Americans, twelve (40%) were Caucasian, three (10%) were Hispanic, two were Asian, one denoted mixed race, and three did not answer the question. Twenty-three participants (77%) used a manual wheelchair. Nineteen (63%) had a spinal cord injury, six (20%) had an amputation, two (7%) had multiple sclerosis, and others included Guillain barre (*n* = 1), traumatic brain injury (*n* = 1), and poliomyelitis (*n* = 1).

For Phase 2 Azure vs. V2 reliability and agreement analysis, seven (three men and four women) with an average age of 29.4 years participated for a total of 70 transfers. Four individuals were Caucasian, two were Asian, and one was Hispanic.

The analysis comparing the ML predicted TAI scores between the Azure and V2 involved three participants (1 man and 2 women) with an average age of 32.7 years. They conducted 10 transfers for each of the five different transfer types (150 trials total). One participant was Asian, and the other two were Caucasian.

### 2.2. Equipment

In both phases of the study, participants transferred from a wheelchair to a height-adjustable firm surface (transfer bench). Custom software programs were developed that utilized existing SDKs for each sensor, allowing for visualization and recording of body tracking data. For the V2, a graphical user interface (GUI) was programmed in C#, Visual Studio 2012, .NET Framework 4.0 and used the Windows Kinect SDK for obtaining the joint center coordinates. The RealSense D435 utilized a third-party SDK by Nuitrack, released by 3DiVi Inc., to obtain the coordinates. A custom Python GUI was created to facilitate data collection from the Kinect Azure, utilizing the pyKinectAzure library. Each individual sensor was connected to its own laptop computer for the data collection. The joint center data from all sensors were collected at 30 Hz and saved in *csv (comma-separated value) files. The data were synchronized via timestamps. Specifications for each sensor are shown in Table 1.

### 2.3. Experimental Setup

Each depth sensor was positioned in front of the transfer surface and the wheelchair at a distance specified by the manufacturer (Table 1 and Figure 2).

### 2.4. Data Collection Protocol

#### 2.4.1. Phase 1 RealSense vs. V2

Participants were asked to position their wheelchair to the right of a level-height bench (70 cm × 55 cm) based on their transfer preferences (Figure 2 left). Data collection from the sensors was started, then participants were asked to perform a transfer from the wheelchair to the transfer surface using their habitual technique. Data collection was stopped once participants were stable on the transfer surface. The participant then transferred back to their wheelchair to prepare for the next trial. The participant repeated the transfers up to five times in each direction for a maximum total of 10 transfers. Approximately 3 to 5 min of rest time was provided between trials, and they were allowed to take more time to rest if needed.

#### 2.4.2. Phase 2 Azure vs. V2

Participants were asked to position themselves to the right of the level-height bench (same one as Phase 1) and prepare to perform a transfer (Figure 2 right). They were asked to perform proper technique in accordance with the TAI (e.g., locking the brakes, removing arm rests, positioning the wheelchair at the proper angle, placing feet on the ground, using proper hand grips and arm positioning, flexing their trunk forward during the lift, and landing smoothly). The quality of the SPT was closely monitored by the research staff visually; if the SPT was not performed correctly, the participant was asked to perform an additional transfer. Time was given in between transfers to rest. A total of 10 transfers from the wheelchair to the bench were collected for each participant. Afterward, the participants were asked to perform transfers using 4 different improper transfer techniques (not leaning forward with the trunk, not placing feet on the ground, using a fist instead of an open hand grip, and positioning the leading arm too far out/away from the body). Each type of improper transfer was performed 10 times each. Rest time was provided in between transfers.

### 2.5. Data Processing

The specific joint centers identified by each sensor type are shown in Figure 3 [28,29,30,31]. Time-series joint center data (X, Y, and Z coordinates) were further analyzed using a custom MATLAB script. This analysis was conducted on the raw data from the sensors with no filtering. Filtering was only used to determine the different phases of the transfer (see below). Four key kinematic features important for describing wheelchair transfer biomechanics [4] and detecting differences in TAI proper and improper technique [11] were used to assess the reliability and agreement between the sensor types. All of these features were calculated from the “lift phase” portion of the transfer [4]. The lift phase was determined in the same manner for all sensors and in all phases by analyzing the *x*-axis position of the spine base/waist/pelvis joint center. The raw positional data was filtered with a digital lowpass Butterworth filter with a cutoff frequency of 15 Hz, and the start and end points of the lift phases were determined. The start of the lift phase was determined by the end of the first plateau region, as shown in Figure 4; the lift phase was the positive linear portion; and the landing phase was the last plateau section. These points were determined by finding the last minimum data point and first maximum data point. The four features included: displacement of the spine base/waist/pelvis joint center (DSWP), average plane of elevation angle on the leading side shoulder (LPOE), average elevation angle on the leading side shoulder (LE), and average trunk flexion angle (TF). These features, how they are calculated, and their relevant joint centers are shown in Table 2 and explained below.

DSWP—the displacement of the spine base/waist/pelvis joint center along the *x*-axis, calculated by taking the final position and subtracting the initial position. This value is given in millimeters and converted to centimeters.LPOE—the average plane of elevation angle on the leading side shoulder (the left side). This angle is calculated between two vectors: a normal vector to the chest and the upper arm (left shoulder to elbow), projected onto the transverse plane. The normal vector is calculated by taking the cross product of the trunk vector (e.g., Azure pelvis to upper spine marker) and the shoulder across vector (left to right shoulder markers). This value is given in degrees.LE—the average elevation angle on the leading side shoulder (the left side). This angle is calculated between two vectors: the trunk vector and the upper arm vector. This value is given in degrees.TF—the average flexion angle of the trunk calculated as the angle between the trunk vector and the vertical *y*-axis. This value is given in degrees.

A secondary goal of the study was to determine the agreement between TAI scores auto-generated with the V2 to those auto-generated with the Azure for proper and improper SPTs. Specific details concerning the development of the ML models used to predict (auto-generate) TAI scores can be found elsewhere [11]. These models use features that are generated from the V2 joint center data to predict if each TAI item is performed correctly or incorrectly. As shown in Figure 3, the joint center locations are slightly different between the two sensors. For the key feature calculations, the marker anatomical locations between the two sensors aligned well, with the exception of the SpineShoulder marker on the V2. Several different approximations were attempted, but the one that provided the most similar location to the V2 SpineShoulder marker was by taking the midpoint between the two Clavicle markers on the Azure map. This new approximated marker was named SpineUpper.

The features used in the ML models (e.g., joint angles, joint center displacements, velocities, accelerations, etc.) were then computed using the identified and approximated Azure marker locations and separately using the equivalent V2 marker locations for each transfer. The features generated from each of the sensors were entered into the ML models to predict the TAI items scores. The TAI contains 15 items that break down the transfer into components that are scored either a ‘1’ for properly executed technique or a ‘0’ for improperly executed technique [32]. Only 11 items have been modeled to date using ML methods (Items 1 and 2 and Items 7–15).

### 2.6. Statistical Analysis

All statistical analyses were performed in SPSS 27. Intra-rater reliability and reliability within a sensor were calculated using ICC 3,1 (two-way mixed-effects model with absolute agreement) for all the trials. Inter-rater reliability, reliability between sensors, was calculated using ICC 2,1 (two-way random-effects model with absolute agreement). ICC values for both analyses were characterized as excellent (ICC > 0.8), good (ICC 0.6–0.79), moderate (ICC 0.4–0.59), fair (ICC 0.2–0.39), or poor (ICC < 0.2). Furthermore, agreement between sensor features was evaluated using Bland–Altman plots. The “agreement” in this study is the characteristic that describes how close the two measurements are. The *infimum* (inf) and *supremum* (sub) of the agreement were set as:infinium=m−1.96×s
supremum=m+1.96×s
where “*m*” is the mean of differences of both sensors and “*s*” is the standard deviation. Based on the Gaussian hypothesis, if 95% of the data are within the range between the inf and sub, it is valid to affirm that the two methods are interchangeable [33].

A percent agreement was computed between the Azure, and V2 ML predicted TAI scores across 150 transfers (3 participants × 50 trials). A confusion matrix was used to determine the accuracy of the ML-predicted TAI scores for each sensor in comparison to the ground truth scores for each transfer.

## 3. Results

### 3.1. RealSense vs. V2 Reliability and Agreement

The V2 showed higher intra-rater reliability (ICC = 0.60–0.82) than the RealSense (ICC = 0.25–0.70) for the four key variables (Table 3). Both V2 and RealSense have high reliability for the TF (ICC = 0.75 and 0.70). The DSWP has the largest difference in ICCs between the two sensors (difference = 0.57).

The LPOE and TF have moderate inter-rater reliability (ICC = 0.52 and 0.51) (Table 4), while the DSWP and LE showed low reliability (ICC = 0.13 and 0.17). The Bland–Altman plots for the agreement between sensors are shown in Figure 5. These show general agreement between the sensors for the recorded trials. Most of the data points fall between the infinium and the supremum for the four key variables, with a few outliers for each. The number of outliers is relatively small compared to the total number of trials (150). Lastly, since there is a general random distribution of the data points around the mean, it is evident that there is no bias of one sensor over the other.

### 3.2. Azure vs. V2 Reliability and Agreement

The ICC values for the Azure were higher (0.91–0.92) than those of the V2 (0.82–0.92) for three of the key variables analyzed (Table 5). The TF angle had approximately the same level of intra-rater reliability as the V2. Additionally, the Azure had a smaller 95% CI range compared to the V2. ICC values were all characterized as excellent. The largest difference in ICC was found in the LPOE angle; however, all differences were less than 0.1.

The DSWP had excellent between-sensor reliability (ICC = 0.91) and all other variables (LPOE, LE, and TF) had good reliability (ICCs = 0.67, 0.63, 0.75) (Table 6). The Bland–Altman plots for the agreement between sensors are shown in Figure 6. These plots show high agreement between the trials between the two sensors. There were a few data points outside of the upper and lower bounds; however, this is a relatively small amount compared to the 70 total trials. With a random distribution of data points around the mean, there did not appear to be bias associated with the errors.

### 3.3. ML Predicted TAI Scores for the Azure and V2

The percent agreement between the two sensors for the predicted TAI scores across all the trials and items was 85.9% (STD 12.8%) (Table 7). The TAI items with the lowest agreement were Items 7 and 11. These items are the position of the feet during the transfer and the hand grip of the leading arm during the transfer, respectively. The items with the highest agreement were Items 2, 14, and 15, related to the angle between the wheelchair and transfer surface, if the transfer was performed in one smooth motion, and if the landing was smooth, respectively.

Both the V2 and Azure sensors showed a high level of accuracy (>77%) for predicting the TAI scores of ‘properly’ executed transfers. Neither of the sensors did well at detecting when improper transfer technique was being used (Table 8). For example, for the transfer that involved not placing the feet on the ground (Item 7), the V2 only detected the issue for 10% of the transfers (3 out of 30 transfers), and the Azure detected it for 7 of the 30 transfers. The V2 did a better job at detecting when the arm was placed in the wrong location (66.7%) compared to the Azure (20%). The Azure did better at predicting when the trunk was not leaned forward enough (40.0%) versus 0% for the V2.

## 4. Discussion

A recent application was developed to auto-score the quality of wheelchair transfer techniques (TransKinect); however, the sensor that is central to capturing the movement strategies (Microsoft Kinect V2) is no longer being produced. A comparative study was conducted to determine if another sensor could be interchanged with the obsolete one. Overall, the results show that Microsoft’s successor to V2, the Azure, showed even better reliability compared to the V2 and Intel’s RealSense for measuring the key variables important for evaluating the quality of transfer techniques. The Azure also showed high levels of agreement with the V2 in relation to the magnitude values of these variables, and the two closely agreed with each other on the auto-generated scoring of proper and improper transfers. However, when looking at how each of these sensors did in detecting proper from improper technique compared to ground truth, both sensors were good at predicting when ‘proper’ technique was used, but both fell short in predicting when improper technique occurred. These results suggest that the ML models of the TransKinect application may need to be further examined, refined, or retrained to improve their accuracy in predicting improper technique. The results of this study provide important information for the next steps of TransKinect application development.

The quality of wheelchair transfer techniques can be evaluated by multiple components related to wheelchair setup, body setup, and flight movements using the latest version of the TAI 4.0 [8]. The kinematic variables related to the upper extremities and trunk motion have a high correlation with joint force and moment at the shoulder, elbow, and wrist during the transfer [4,34]. Learning to align the upper limb and trunk motions with TAI principles can reduce upper limb joint forces and moments [10]. In our previous studies, we applied kinematic variables as the features of machine learning classifiers to predict TAI scores [11]. The DSWP is a feature that is important for identifying if the person uses the correct wheelchair distance between transfer surfaces (TAI 4.0 item 1), scoots forward to the edge of the sitting area before transfer (item 8), and has good body balance during flight (item 14, 15). The DSWP is also useful for detecting when the lift phase of the transfer process occurs, which is required for the application of the TAI prediction classifiers. The LPOE and LE are important for identifying the correct technique related to the positioning of the arm (items 9, 11, 12). The TF is a key feature for measuring correct trunk leaning motion (item 13). Using this movement pattern can reduce the upper extremities joint loading during transfer [4,8,11].

We examined the intra-rater reliability of the sensors by analyzing repeated trials performed by the same participant. Some intra-trial variability is expected because the individuals could have varied their technique for each transfer. However, because both sensors are watching the same transfers, the variation from the participants would have been controlled for in the analysis. Thus, the intra-rater reliability from the repeated transfers provides an indication of how reliable the sensors are relative to each other in detecting the motions.

The reliability analysis comparing RealSense to the V2 shows that the RealSense would not be a viable substitute for the V2 in the TransKinect application. The RealSense had worse intra-sensor reliability than the V2, with only two of the key variables scoring a good ICC (the other two being fair). On the other hand, the V2 had two excellent and two good ICC values. Additionally, the RealSense had broader 95% confidence intervals than the V2. Furthermore, the inter-sensor reliability analysis showed that the RealSense and V2 have moderate ICC for only two of the key variables (LPOE and TF). The sensors had a very low agreement for DWBW and LE, the former being the variable used to identify the different phases of the transfer. These results indicate that the V2 is much more reliable for repeated data collection than the RealSense, and that the RealSense should not be substituted into the TransKinect application due to its lower intra- and inter-sensor reliability than the V2.

The reliability analysis between the V2 and the Azure, however, showed an opposite trend. While all the key variables showed excellent intra-sensor reliability, the ICC values for the Azure were even higher than those of the V2. The 95% confidence intervals for the Azure were also narrower than the V2, indicating greater stability of the calculated features. The inter-sensor reliability between the V2 and the Azure also showed ICC values in the good and excellent range. The Bland–Altman plots also showed that the magnitude differences between the two sensors for the key variables are small and that most of the data lie within the 95% limits of agreement with the exception of a few outliers. These results suggest that the Azure could be a viable substitute for the V2 in the TransKinect application.

Before full integration of the Azure into the TransKinect application, a proper validation and compatibility test of the ML models used to predict the TAI item scores is necessary. The results showed that the Azure and V2 had high levels of agreement on most TAI items, indicating that the Azure should be compatible with the existing ML models. All but two items scored an agreement over 75%, Item 7 and 11 (59.3% and 71.3%) (feet position and leading arm-hand grip). It should be noted that the V2 has very low accuracy in body tracking with the lower extremities for individuals in wheelchairs. Often, the sensor would confuse the wheelchair frame or headrest as part of a body and interfere with the data collection. It was specifically seen in this analysis that the pelvis marker on the V2 would consistently lag behind the actual body; additionally, there were times when the V2 lost sight of several segments during the transfer. This problem was not seen in the Azure data collection; all joints were tracked accurately, and there was no data loss for any of the joints. These differences between the actual body position and the skeletons of the two sensors are shown in Figure 7. However, while the two sensors agreed well with each other on the ML-predicted TAI scores for most items, we felt it was also important to assess how their predicted scores compared to ground truth.

The accuracy of the ML-predicted TAI scores using both Azure and V2 sensor data tested against the ground truth scores for proper and improper transfer techniques showed high accuracy in detecting proper technique when proper technique was used but tended to assign proper technique when improper technique was used. Thus, the current ML models appear to over predict good quality transfers, regardless of the sensor used. There are several possible reasons this could have occurred. The ML models were developed using 100 unique wheelchair users who each contributed multiple transfer trials (up to five) to the analysis. All the trials were randomized and split into a 70/30 training and testing set. It is possible that some of the same subject’s trials were part of both the training and testing sets. While this is a relatively common practice in ML, this method can lead to overfitting [35]. Therefore, while the original ML validation showed moderate to high accuracy for the prediction of TAI items, they are seeing ‘new’ data in this study that may look different than that contained in the original dataset. Furthermore, the initial training and testing dataset had more proper than improper technique data. To overcome these obstacles in the future, the ML models should be retrained, ideally with the Azure, as it is a more reliable and robust sensor compared to the V2, with a proper balance of proper and improper transfer technique.

## 5. Conclusions

The TransKinect application offers a promising opportunity to automate the evaluation of SPTs and support clinical decision making. The use of commercial hardware, beginning with the low cost, marker-less Kinect V2 sensor, makes the system more accessible to clinics and easier to use. However, using commercial hardware may also lead to the inevitable possibility of discontinued production leading to the need to find an acceptable substitute depth sensor for future use. The Intel RealSense, while an affordable commercially available device, unfortunately, does not demonstrate the necessary reliability and agreement with the V2 to justify it as a viable substitute sensor. On the other hand, the Azure demonstrates sufficiently high inter- and intra- rater reliability when compared to the V2, which suggests it may be an adequate replacement within the TransKinect system. The final, and most clinically relevant, finding from this analysis is the true accuracy of the ML algorithms used to produce TAI scores from the collected data. The results of this analysis suggest that while the Azure and Kinect V2 are tracking body movement throughout the transfer process precisely, the predicted ML TAI scores did not match the ground truth scores for transfers that were performed using improper technique.

The future steps for this application are to retrain the ML models with the new Azure sensor, placing an emphasis on identifying incorrect technique, using a more balanced dataset, and performing testing validation on a new batch of subjects.

## Figures and Tables

**Figure 1 sensors-22-04977-f001:**
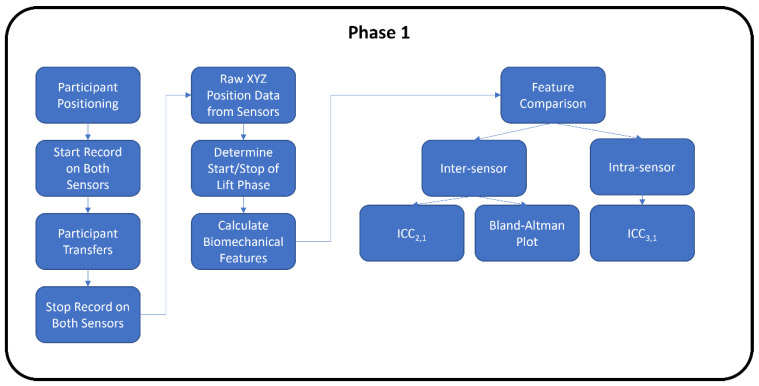
Experimental study flowchart for Phase 1 and Phase 2.

**Figure 2 sensors-22-04977-f002:**
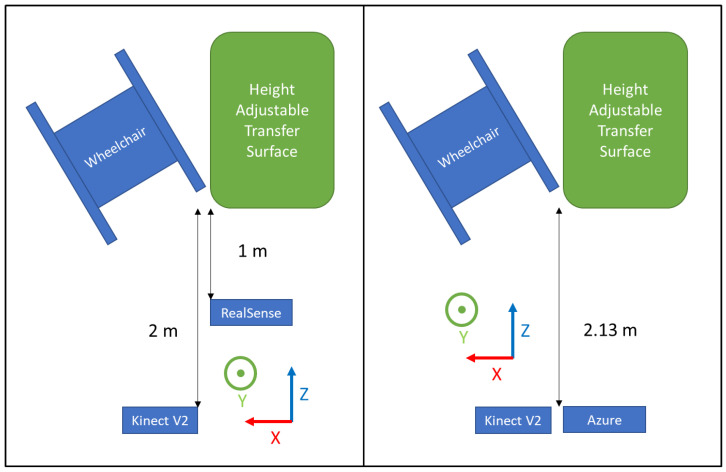
Experimental setup for Phase 1 (**left**) and Phase 2 (**right**).

**Figure 3 sensors-22-04977-f003:**
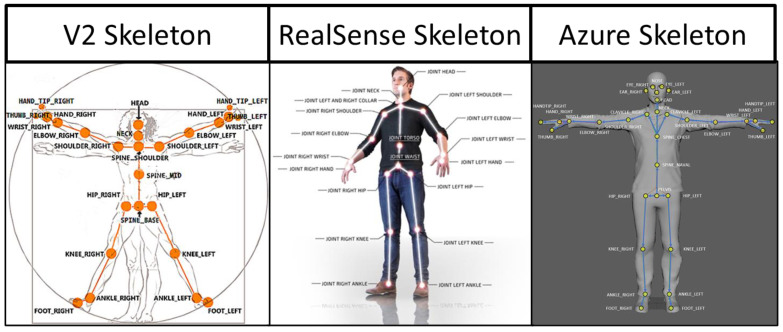
Skeletal joint center maps for each sensor.

**Figure 4 sensors-22-04977-f004:**
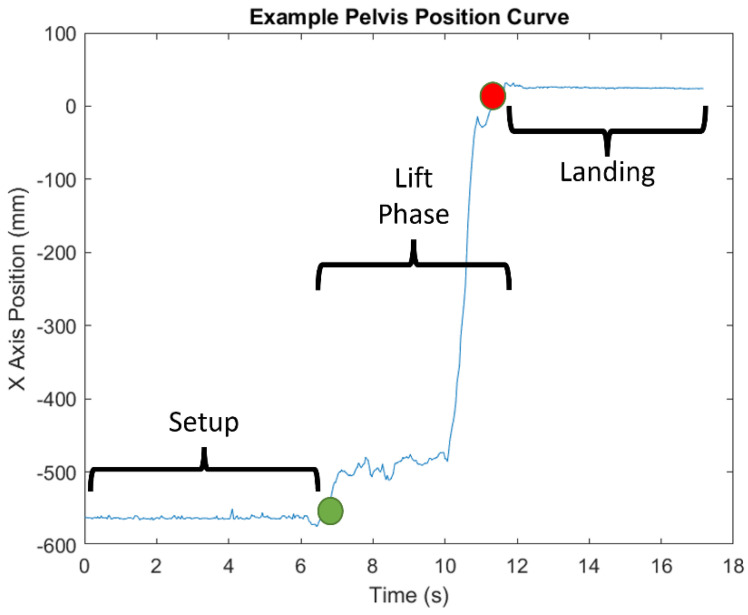
Example of *x*-axis position of the pelvis marker and the different phases of transfer.

**Figure 5 sensors-22-04977-f005:**
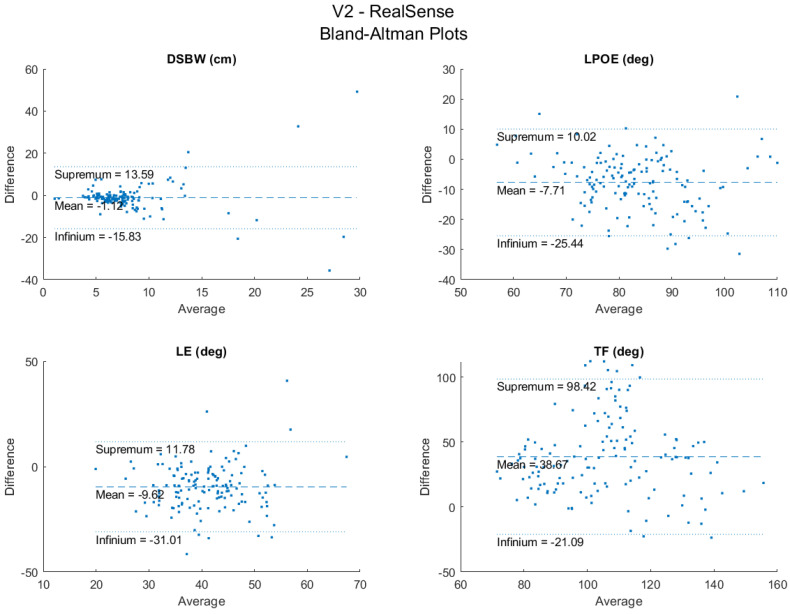
Bland−Altman plots between RealSense and V2.

**Figure 6 sensors-22-04977-f006:**
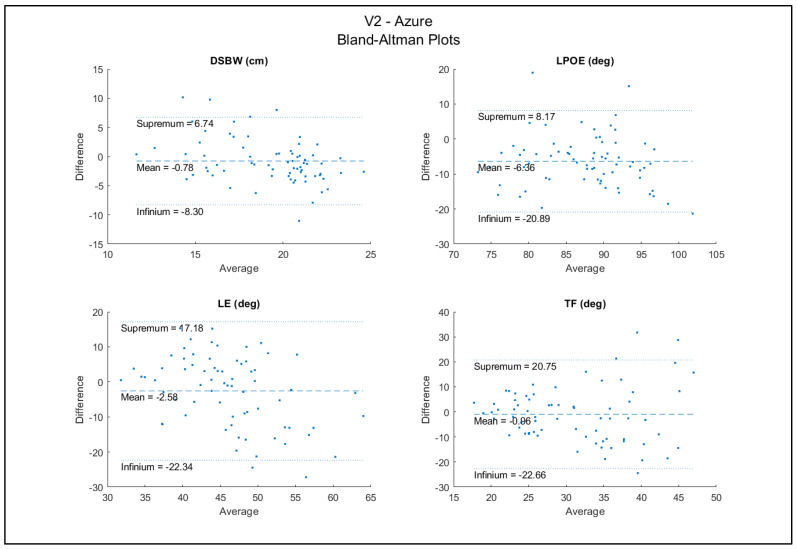
Bland−Altman plots between Azure and V2.

**Figure 7 sensors-22-04977-f007:**
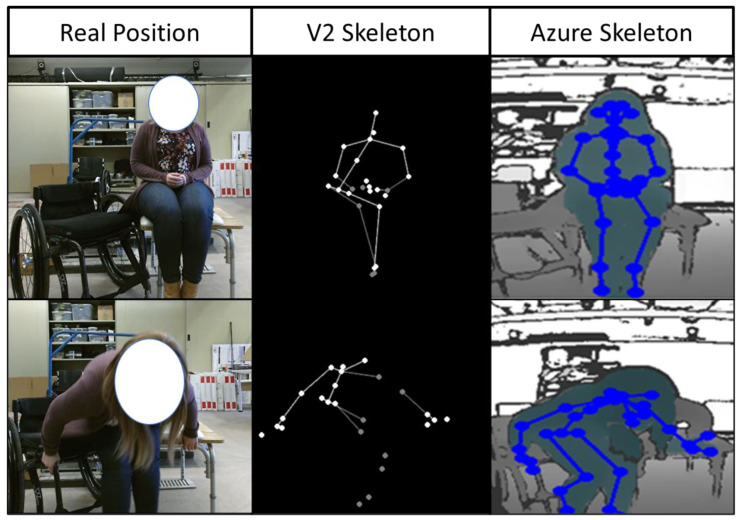
Skeletal approximation differences between Azure and V2 sensors compared to the real body positioning.

**Table 1 sensors-22-04977-t001:** Hardware specifications of depth sensors.

Sensor	Depth Camera Resolution (pixels)	Ideal Operating Range (m)	Sampling Frequency (Hz)	Overall Dimensions (mm)	Images
**Kinect** **V2**	512 × 424	0.5–4.5	≤30	249 × 66 × 67	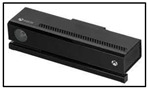
**Kinect** **Azure**	640 × 576	0.5–4	≤30	103 × 39 × 126	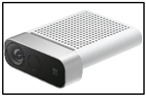
**Intel** **RealSense**	1280 × 720	0.3–3	≤90	90 × 25 × 25	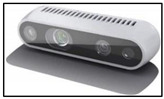

**Table 2 sensors-22-04977-t002:** Key variables used in analysis of sensor reliability and agreement.

Feature	Description	Relevant Joint Centers	
**DSWP**	Displacement of the SpineBase, waist, or pelvis along the *x*-axis, measured in millimeters.	V2	SpineBase
		RealSense	Waist
		Azure	Pelvis
**LPOE**	Average joint angle between a normal vector orthogonal to the trunk and shoulder vector projected onto the transverse plane measured in degrees.	V2	SpineBase, SpineShoulder, LeftShoulder, RightShoulder, LeftElbow
		RealSense	Waist, left and right collar, LeftShoulder, RightShoulder, LeftElbow
		Azure	Pelvis, SpineUpper *, ShoulderLeft, ShoulderRight, ElbowLeft
**LE**	Average joint angle between the trunk and the upper arm, measured in degrees.	V2	SpineBase, SpineShoulder, LeftShoulder, LeftElbow
		RealSense	Waist, left and right collar, LeftShoulder, LeftElbow
		Azure	Pelvis, SpineUpper *, ShoulderLeft, ElbowLeft
**TF**	Average joint angle between the trunk and the vertical *y* −axis, measured in degrees.	V2	SpineBase, SpineShoulder
		RealSense	Waist, left and right collar
		Azure	Pelvis, SpineUpper *

* The SpineUpper joint center on the Azure was approximated to match the SpineShoulder on the V2. This was completed by taking the midpoint of the two collarbone markers.

**Table 3 sensors-22-04977-t003:** Intra-rater reliability ICC 3,1 within RealSense and V2.

	RealSense				V2		
	ICC	Confidence Interval Lower Bound	Confidence Interval Upper Bound	ICC	Confidence Interval Lower Bound	Confidence Interval Upper Bound	Diff.
**DSWP**	0.25	0.11	0.44	0.82	0.72	0.90	0.57
**LPOE**	0.60	0.44	0.75	0.81	0.71	0.89	0.21
**LE**	0.38	0.22	0.57	0.60	0.44	0.75	0.22
**TF**	0.70	0.56	0.82	0.75	0.63	0.85	0.05

**Table 4 sensors-22-04977-t004:** Inter-rater reliability ICC 2,1 between RealSense and V2.

	ICC	Confidence Interval Lower Bound	Confidence Interval Upper Bound		Mean	Std
**DSWP (cm)**	0.25	0.11	0.55	**RealSense**	75.11	59.59
				**V2**	86.31	54.18
**LPOE (deg)**	0.57	0.07	0.84	**RealSense**	79.28	9.96
				**V2**	86.99	11.51
**LE (deg)**	0.13	0.10	0.40	**RealSense**	36.15	9.27
				**V2**	45.77	8.67
**TF (deg)**	0.63	0.06	0.85	**RealSense**	21.86	8.22
				**V2**	27.79	10.51

**Table 5 sensors-22-04977-t005:** Intra-rater reliability ICC 3,1 within Azure and V2.

	Azure			V2			
	ICC	Confidence Interval Lower Bound	Confidence Interval Upper Bound	ICC	Confidence Interval Lower Bound	Confidence Interval Upper Bound	Diff.
**DSWP**	0.92	0.79	0.98	0.84	0.58	0.97	0.08
**LPOE**	0.92	0.78	0.98	0.82	0.54	0.96	0.09
**LE**	0.92	0.79	0.98	0.89	0.71	0.98	0.03
**TF**	0.92	0.78	0.98	0.92	0.79	0.98	0.01

**Table 6 sensors-22-04977-t006:** Inter-rater reliability ICC 2,1 between Azure and V2.

	ICC	Confidence Interval Lower Bound	Confidence Interval Upper Bound		Mean	Std
**DSWP (cm)**	0.91	0.51	0.98	**Azure**	47.75	5.38
				**V2**	49.76	6.63
**LPOE (deg)**	0.67	−0.16	0.94	**Azure**	84.38	5.49
				**V2**	90.74	4.92
**LE (deg)**	0.63	−0.89	0.94	**Azure**	44.83	5.20
				**V2**	47.41	7.29
**TF (deg)**	0.75	−0.67	0.96	**Azure**	30.58	7.35
				**V2**	31.54	7.49

**Table 7 sensors-22-04977-t007:** Agreement between Azure and V2 ML outcomes from 150 transfers.

TAI Items	Description	AZ == V2	AZ =/= V2	Percent Agreement
**1**	Distance Transferred	135	15	90.0
**2**	Angle of Approach	149	1	99.3
**7**	Feet Position	89	61	59.3
**8**	Scoot Forward	135	15	90.0
**9**	Leading Arm Before Transfer	117	33	78.0
**10**	Push-off Hand Grip	133	17	88.7
**11**	Leading Hand Grip	107	43	71.3
**12**	Leading Arm After Transfer	119	31	79.3
**13**	Trunk Lean	134	16	89.3
**14**	Smooth Transfer	149	1	99.3
**15**	Stable Landing	150	0	100.0
**Average**		128.8	21.2	85.9
**STD**		19.2	19.2	12.8

**Table 8 sensors-22-04977-t008:** Percent accuracy of ML outcomes from Azure (AZ) and V2 compared to ground truth (30 trials of each transfer type). The shaded cells indicate the items that were targeted for each improper transfer type and results.

Improper Transfers										
	Good		Feet		Trunk		Arm		Fist	
**Items**	V2	AZ	V2	AZ	V2	AZ	V2	AZ	V2	AZ
**1**	100.0	93.3	100.0	96.7	100.0	86.7	100.0	90.0	100.0	83.3
**2**	100.0	100.0	100.0	96.7	100.0	100.0	100.0	100.0	100.0	100.0
**7**	90.0	80.0	10.0	23.3	83.3	60.0	93.3	63.3	80.0	73.3
**8**	96.7	100.0	70.0	100.0	93.3	96.7	96.7	100.0	100.0	96.7
**9**	80.0	96.7	86.7	100.0	100.0	100.0	66.7	20.0	80.0	100.0
**10**	96.7	96.7	93.3	93.3	100.0	83.3	96.7	93.3	90.0	100.0
**11**	83.3	83.3	96.7	63.3	73.3	90.0	90.0	100.0	23.3	20.0
**12**	76.7	96.7	80.0	100.0	100.0	100.0	33.3	0.0	76.7	100.0
**13**	100.0	96.7	100.0	93.3	0.0	40.0	100.0	100.0	100.0	96.7
**14**	100.0	100.0	96.7	100.0	100.0	100.0	100.0	100.0	100.0	100.0
**15**	100.0	100.0	100.0	100.0	100.0	100.0	100.0	100.0	100.0	100.0

## Data Availability

The data presented in this study are available upon request from the corresponding author.

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
