# Peer review of "Reliability of 3D Depth Motion Sensors for Capturing Upper Body Motions and Assessing the Quality of Wheelchair Transfers"

_sensors, 2022, doi:10.3390/s22134977_

Round 1

Reviewer 1 Report

Authors evaluate the quality of SPT using various densors in addition to the Kinect and Transkinect. 
Study evaluates the effectiveness of using RealSense and Azure sensors to act as surrogate sensors for transkinect application. 
Also agreement of the TAI score with kinect v2 with the other sensors are evaluated. 

Question to the authors

Were calibrations required while collecting depth data?

Figure 1: Different distances have been used for kinect v2 and realsense.  Shouldnt they be at the same distance for fair comparison.  Although realsense have
much smaller range compared to kinect v2.  Can there be more explaination on why different distances were used?

Figure 1: Similarly, what is the motive to use a distance of 2.13 for kinect v2 and azure? 

151 - Could these improper transfers affect patients?

How well is the approach adaptable for people of different heights

For the features that are calculated, DSWP, LPOE, LE, TF, they do not require using the depth information.  Can this be solved with a much simpler
RGB camera (Inexpensive) and use some deep learning techniques such as OpenPose, AlphaPose to extract keypoints.  Further there are techniques to estimate (x,y,z)
from just RGB images.

There has been similar work in the past that uses body keypoints to evaluate actions performed and correctness of actions performed
with body keypoints.  

1. Dillhoff, A., Tsiakas, K., Babu, A.R., Zakizadehghariehali, M., Buchanan, B., Bell, M., Athitsos, V. and Makedon, F., 2019, 
September. An automated assessment system for embodied cognition in children: from motion data to executive functioning. 
In Proceedings of the 6th international Workshop on Sensor-based Activity Recognition and Interaction (pp. 1-6).

2. Lai, K.T., Hsieh, C.H., Lai, M.F. and Chen, M.S., 2010, June. Human action recognition using key points displacement.
In International Conference on Image and Signal Processing (pp. 439-447). Springer, Berlin, Heidelberg

For the ML model prediction: As the prediction is a countinous value ranging from 0-1, how were the ground truth labelled.  What does a value of 0.4 or 0.7 mean and how were they correctly quantified for ground truth?

From the article, it can be seen that the major contribution is a comparitive study between different image sensors in estimating the body keypoints.  As once the keypoints are estimated, the TAI score calculation is same across
the different sensors.  But a literature study shows that there are existing work that shows the difference in estimating the body keypoints between sensors, 

Kinect™ and Intel RealSense™ D435 comparison: a preliminary study for motion analysis

Comparative study of intel R200, Kinect v2, and primesense RGB-D sensors performance outdoors

Accuracy measurement of depth using Kinect sensor

How is your work different from these?

Minor fixes

Please check grammatical errors in "Data processing section"
Line 216 - 217: Could be reworded. 

Author Response

Please see the attachment, thank you! 

Reviewer 2 Report

Dear Authors,

thank you very much for sending the article titled: Reliability of 3D Depth Motion Sensors for Capturing Upper Body Motions and Assessing the Quality of Wheelchair Transfers. Generally, the paper is quite interesting to my mind, however, the authors should refer to the following statements:

- please include a Flowchart at the beginning of the Materials and Methods section

- line 232,233, standard deviation for population is Sigma, wheras for sample s.

- I recommend moving lines 242-259 to the Materials and Methods section

-  could you include in the Table 5, p-value between Azure and V2?

- Lack of the future studies. For example in this manuscript authors used Kinect for analyze repeated trials performed by the same participant. It is an optical system. In summary, the directions of further research could be presented. For example in the article titled: Body part accelerations evaluation for chosen techniques in martial arts, 2017 authors use IMU for analyze kinematic parameters. It would be useful in your case. To improve quality of the manuscript I suggest cite this article and write a few sentences in Conlcuison section.

Author Response

Please see the attachment, thank you! 
